# Transport stress induces paradoxical increases in airway inflammatory responses in beef stocker cattle

Grace M. Jakes[1☉], Dylan T. Ammons[1☉], Randy Hunter[2☉], Steven Dow[1,3☉]*,
Sarah M. Raabis [3☉]*

**1** Department of Microbiology, Immunology, and Pathology, Colorado State University, Fort Collins, Colorado, United States of America, **2** Hunter Cattle Company, Wheatland, Wyoming, United States of America, **3** Department of Clinical Sciences, Colorado State University, Fort Collins, Colorado, United States of America

☉ These authors contributed equally to this work.
* sdow@colostate.edu (SD); Sarah.Raabis@colostate.edu (SMR)

## Abstract

The development of Bovine Respiratory Disease (BRD) in beef cattle is associated with stressful events, including auction and transport. In addition to the effects of commingling on pathogen exposure, stress also impacts immune function and has classically been associated with an immunosuppressed state. Much of the research on cattle immunity in stress has focused on peripheral blood immune functionality rather than mucosal immune responses. To characterize immunity at the primary site of pathogen colonization in BRD, we evaluated stress responses in lung cells from beef stocker cattle to understand mucosal immune changes. Stocker calves were sampled via bronchoalveolar lavage fluid (BALF) collection within 24 hours of auction and transport to a new facility (Stressed, n = 12), or allowed to acclimate for 2 months at the new facility before sampling (Acclimated, n = 7). Lavage cellular RNA was extracted and sequenced for bulk RNA-seq gene expression. Differential gene expression analysis of RNA sequencing data demonstrated a profound upregulation of inflammatory genes in Stressed calves compared to Acclimated calves, including expression of *CXCL8*, *CSF3R*, *IL1B*, and *CCL22*. Top pathway upregulation in Stressed calves involved neutrophil migration and chemotaxis, and cytokine signaling. To predict cellular population proportions from the data, CIBERSORTx was used to deconvolute bulk RNA-seq gene counts. This analysis showed that Stressed calves had significantly increased BALF neutrophils compared to Acclimated calves (p = 0.003). Neutrophilic infiltration occurred in the absence of pathogen colonization of the lungs in most calves, as demonstrated by a multi-pathogen respiratory qPCR screen. As such, the stress induced an inflammatory response in lungs not explained by pathogen exposure. This study provides strong evidence that shipping stress in beef stocker calves can trigger increased inflammatory pulmonary mucosal immune responses, which has important implications for the pathogenesis of BRD.

**Data availability statement:** The data presented in this paper are available through the Dryad open data publishing platform at https://doi.org/10.5061/dryad.02v6wwqh6. The data discussed in this publication have also been deposited in NCBI's Gene Expression Omnibus and are accessible through GEO Series accession number GSE299548 (https://www.ncbi.nlm.nih.gov/geo/query/acc.cgi?acc=GSE299548). An associated GitHub page containing all the analysis code and software versions used to analyze the data presented in this article is available at (https://github.com/gmjakes/Bovine_bulkseq_stress_study). Any additional data requests can be made by contacting the corresponding authors.

**Funding:** Authors SD and SR were granted a Cooperative Agreement with the USDA Agricultural Research Service (https://www.ars.usda.gov/). FAIN: 58-3022-3-023. The sponsor had no role in the study design, data collection and analysis, decision to publish, or in the preparation of the manuscript.

**Competing interests:** The authors have declared that no competing interests exist.

## Introduction

Bovine respiratory disease (BRD) is the leading cause of morbidity and mortality in beef cattle raised in the United States [1,2]. One of the major risk factors for BRD is the stress of transport and commingling at a new facility, where cases typically peak in the first weeks following arrival [3–5]. Given the fact that transport and commingling are experienced by the vast majority of U.S. beef cattle [1,6], understanding the mechanisms of immune regulation following these stressors is imperative to mitigate disease. Stress has classically been considered immunosuppressive due to the down-modulatory effects of stress-induced glucocorticoids on inflammatory and lymphocyte responses [7]. Glucocorticoids regulate inflammatory responses principally through the downregulation of NFκB-mediated signaling [7], which in turn mitigates the production of proinflammatory cytokines including TNF-α [8], IL-1β [9], IL-6 [10], and IL-8 [11] across species and cell classes. Additionally, glucocorticoids attenuate lymphocyte responses and serve as pro-apoptotic factors to reduce lymphocytes in circulation [12]. In the peripheral blood, stress in cattle has been demonstrated to delay humoral responses to vaccination [13].

However, the effects of shipping stress on mucosal immune responses in the lungs have not been thoroughly evaluated, and it remains possible that pulmonary immune responses may differ from peripheral immune responses in stressed cattle. While stress is thought to decrease lymphocyte functionality [5], immune responses are not always attenuated in cattle following stress. For example, transit stress is known to increase cellular recruitment to the lung following LPS challenge in cattle [14], and additive stressors such as weaning and transport can result in greater systemic inflammatory responses and increased mortality in experimental infection [15]. Cattle in the first weeks following arrival at a new facility can express higher levels of inflammatory cytokines in the peripheral blood [16], and in response to stimulation [17]. However, it is unclear if this is in response to immunosuppression and greater pathogen colonization in the lower airway, or if there are other mechanisms where stress can upregulate inflammatory and cellular recruitment responses. Increasingly, studies in humans and mice have demonstrated that acute and chronic stressors can upregulate systemic inflammatory responses in the absence of overt pathogen challenge [18–21], and that this may in turn affect mucosal tissues such as the gastrointestinal tract and lung.

New insights into immune regulation in stressed cattle can be limited by a number of factors, one of the greatest of which is the lack of reagents for cell analysis by flow cytometry or for cytokine analysis by ELISA. Recently, with the advent of bulk transcriptomic mRNA sequencing (RNA-seq) technologies, global immune regulation can be evaluated to determine host responses to physiologic stressors. In cattle, RNA-seq has been used to predict gene expression patterns in the peripheral blood which are associated with increased risk of BRD development [22]. Other studies have demonstrated that modulation of lung immune responses in experimental infection can result in improved calf morbidity and mortality outcomes [23]. These studies have highlighted that an excess of inflammatory immune responses is often linked with more severe morbidity.

In the current study, we sought to increase our understanding of mucosal immune responses in the lungs following shipping stress by leveraging RNA-seq to determine pathways of immune regulation. Our hypothesis was that transported calves would experience an increase in systemic stress markers including cortisol, haptoglobin, and serum amyloid A, and that adaptive immune responses would be attenuated at the mucosal surface in the lungs following shipping stress, resulting in an increase in pathogen recovery in the lower airway. The findings reported here provide unexpected results regarding the nature of mucosal immune regulation in cattle following auction and transport stress and demonstrate that inflammation rather than immunosuppression predominates in calves after stress, even in the absence of detectable pathogen colonization of the lower airway. These findings provide important context for the understanding of immune responses leading to BRD risk in cattle.

## Materials and methods

### Study population

Beef calves (5–6 months of age) were comingled and purchased at auction and transported 228 miles to a commercial backgrounding operation (Wheatland, WY). The first group of calves was sampled within 24 hours of arrival at the backgrounding facility before vaccination and branding (Stressed), and the second group of calves was sampled after 2 months at the facility (Acclimated, Fig 1). Calves arrived at the backgrounder in separate shipments, allowing for random sampling throughout the backgrounding period. A power analysis was conducted based on expected differences in systemic acute phase protein levels, as they are validated markers associated with immune activation and the stress response. The power analysis was conducted in R using the pwr package [24] for a two-sample t-test with a Cohen's d of 1.4, expected 5-fold difference in mean levels of acute phase proteins (including lipopolysaccharide binding protein (LBP), serum amyloid A (SAA), and haptoglobin), and significance set to 0.05 [25]. This yielded a desired sample size of 9 per group. To account for expected proportions of calves that would go on to develop BRD from the Stressed group, additional Stressed calves were sampled for a total of 12 Stressed calves. Calves were not administered metaphylactic antibiotics at any time during the study. All calves were administered the following vaccinations according to label instructions: Pyramid 3 + Presponse SQ® (Bovine Viral Diarrhea Virus types 1 and 2, Bovine Herpes Virus 1 and *Mannheimia Haemolytica* toxoid, BI Animal Health), as well as Vision 7® (7-way *Clostridium* spp.vaccine, Merck Animal Health) at receiving. Calves were managed on pasture for the duration of the backgrounding period (approximately 7 months), and health evaluation was conducted by pen riders. All procedures in the study were approved by the Colorado State University Institutional Animal Care and Use Committee (IACUC protocol # 5749), with all BALF collected using veterinarian-administered xylazine (as described in the Sample Acquisition and Health Screening section), and every effort made to reduce animal suffering.

### Sample acquisition and health screening

Calves were handled according to humane best handling practices, and all sampling and health screening was performed using a Silencer® chute. Bronchoalveolar lavage fluid was collected from healthy calves. Exclusion criteria included an elevated rectal temperature (>39.7 °C), at least 1 cm$^2$ of lung consolidation on ultrasound, or clinical signs consistent with BRD, including spontaneous cough, and moderate to severe nasal or ocular discharge [26]. Lung ultrasound was performed with an Ibex Pro® 5–9 MHz rectal linear probe (E.I. Medical Imaging®, Loveland, CO) by author SR, as previously described [27], with the exception that the right cranial lung lobe could not be consistently evaluated due to limitations of the chute design. Due to logistical reasons, only 7 calves were sampled from the Acclimated group. BALF sampling was conducted in standing animals sedated with xylazine (20 mg/ml Ana-Sed®) at 0.01 mg/kg IV. A 5 Fr red rubber catheter (JorVet™) was inserted into both nares to apply 100–160 mg lidocaine (20 mg/ml VetOne®) to the arytenoids. A sterile cuffed bronchoalveolar lavage catheter (MILA® BAL 240 cm) was passed through the ventral meatus, arytenoids and into the trachea with visualization using a modified video endoscope (S.E.C Repro®). When the catheter reached a terminal

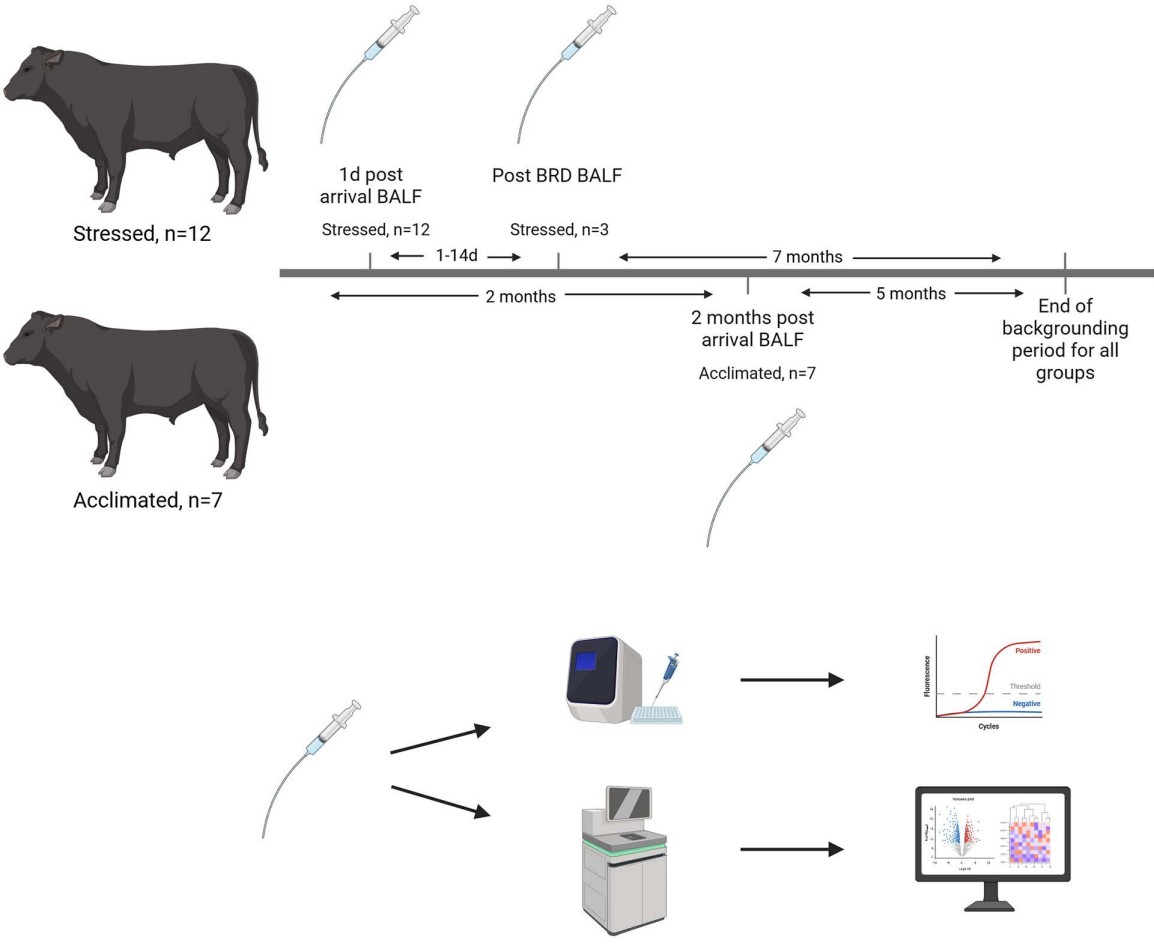

**Fig 1. Study design.** Healthy stressed calves (n = 12) were sampled within 24 hours of arrival at the backgrounding facility. Healthy acclimated calves (n = 7) were likewise sampled after 2 months at the backgrounding facility and BALF was processed for sequencing analysis. Stressed calves who experienced BRD signs in the first 14 days at the facility were sampled a second time just prior to first BRD treatment to determine the etiology of infection. Figure created using Biorender.

bronchus, a 30 cc balloon was inflated and 120 ml of sterile saline was instilled prior to removing the BALF sample. BALF samples were immediately placed on ice until processing at the laboratory (approx. 4 hours). Following BALF collection, peripheral blood was collected for serum analysis (haptoglobin, SAA, LBP, cortisol) via jugular venipuncture and placed on ice until processing at the laboratory.

Acclimated calves were evaluated for BRD signs including reduced rumen fill, nasal discharge, and depressed attitude before sampling and after sampling by trained pen riders. Acclimated calves were excluded from the study if they were ever pulled for BRD treatment prior to the 2-month sampling timepoint. Calves in the acclimated group were never pulled by pen riders for clinical signs consistent with BRD. Stressed calves were likewise evaluated for BRD by trained pen riders after sample collection for the remainder of the backgrounding period. During the study, several Stressed calves developed BRD signs in the first two weeks after arrival. Any previously sampled calf displaying BRD signs was pulled and screened for elevated rectal temperature >39.7 °C and lung consolidation on ultrasound. Calves meeting either of these criteria were then sampled for BALF and treated with tulathromycin and ketoprofen intramuscular injection (Draxxin® KP, Zoetis, Parsippany-Troy Hills, NJ). Two stressed calves experienced BRD signs more than 6 weeks after arrival to the

facility and could not be sampled for BALF due to technical reasons. BALF samples were handled as above and screened for major BRD pathogens by qPCR (as below). Stressed calves who remained healthy after arrival were not pulled for sampling again subsequent to the initial sampling timepoint. For the purpose of downstream RNA-seq analysis, calves were classified as having developed BRD if signs were noted within the first two weeks at the facility.

## Sample processing

Prior to filtering and centrifugation, a ~ 2 ml aliquot of BALF was saved for qPCR analysis for major BRD pathogens. BALF samples were filtered through a sterile 70 µm filter and immune cells were pelleted by centrifugation at 500 x g for 10 minutes. BALF supernatant was removed, and RNA was extracted from cells using the RNeasy Mini Kit (Qiagen, Hilden Germany). Blood was allowed to clot and then centrifuged at 3000 x g for 5 minutes. Serum was separated and stored at −80 °C until further analysis.

## qPCR analysis of BALF

BALF samples were analyzed at the Iowa State University Veterinary Diagnostic Laboratory via qPCR for BRD pathogens, including *Mannheimia haemolytica*, *Pasteurella multocida*, *Histophilus somni*, *Mycoplasma bovis*, Bovine Herpes Virus 1, Bovine Viral Diarrhea Virus, Bovine Respiratory Syncytial Virus, and Bovine Coronavirus. Threshold cycle (Ct) values were collected for each pathogen, with the positivity threshold set to <35. For animals who later went on to develop BRD, a second BALF sample was collected at the time of diagnosis and screened for pathogens using the same qPCR panel.

## Serum cortisol and acute phase protein assessment

Serum haptoglobin, serum LBP, SAA, and serum cortisol levels were assessed using commercially available kits (hapto-globin: Immunology Consultants Laboratory, Inc. E-10HPT; LBP: Hycult Biotech; SAA: Tridelta Development Ltd.; cortisol: Seimens Diagnostics) according to manufacturer's instructions. All samples and standards were plated in duplicate.

## Sequencing

RNA libraries were sequenced by Novogene Corp. using an Illumina Novaseq 6000 platform (Novogene Co., Sacramento, CA). RNA quality was evaluated using an Agilent 2100 Bioanalyzer system to verify minimum RNA integrity levels.

Libraries were sequenced on an Illumina PE150 (Novaseq) platform for 40M raw reads per sample. Raw data were filtered using Fastp (version 0.23.2) by removing reads containing adapters and for Phred scores >30. The filtered reads were then aligned with STAR (version 2.7.10b), ARS-UCD1.3 genome assembly. Aligned reads were annotated and counted using FeatureCounts (version 2.0.1), and differentially expressed genes were identified using DEseq2. Further biological interpretations including gene ontology enrichment and gene set enrichment analysis were then performed using clusterProfiler [28].

## CIBERSORTx analysis: cell identity and proportion prediction

To predict BALF cell population proportions, the CIBERSORTx software was used to deconvolute bulk RNA-seq gene counts. Briefly, a prior bovine BALF single-cell dataset was annotated and used as the "ground-truth" reference for the bulk gene counts expressed in the present study. Five of the Acclimated cattle had paired single-cell and bulk RNA-seq data, which were used as a reference for deconvolution, and to evaluate the accuracy of deconvolution. To predict cellular proportions, a gene count matrix including identity assignment of major immune cell subsets was imported into the CIBERSORTx software. Raw gene counts were imported into the software from Stressed and Acclimated calves, and ν-support vector regression with constraints was performed to find the best set of cell fractions whose weighted combination of gene expression best explained the bulk data. Single-cell reference uniform manifold approximation and projection (UMAP), as well as visual representation of cell population marker expression can be found in S1 Fig. Cell identity

                                                                 

assignment for the single-cell reference was performed using classical surface marker and transcription factors for each major cell population. Table 1 contains a select representation of key markers used to define each cellular population. A complete list of up and down regulated markers (genes) defining each major cell cluster included in the CIBERSORTx matrix can be found in S1 Table.

Two batch correction methods (noBatch, bBatch), which account for the bias of cell population prediction based on the single-cell assay type used, were evaluated. Percent similarity matrices for each batch correction method were used to determine the best batch correction method for this dataset, and to evaluate the accuracy of the cellular proportion prediction (S2 Fig). Correlation $R^2$ values were calculated between the "ground truth" single-cell references and the bulk sequencing data and can be found in S3 Fig. Based on these two analyses, we elected to use the bBatch correction method to account for assay-induced single-cell population proportion prediction bias.

### Statistical analysis

Statistical analyses for serum cortisol, acute phase proteins, clinical data, and an odds ratio were completed with Prism8 software (GraphPad) using the Mann-Whitney U test or a Student's t-test and the Fisher's exact test, respectively. The results are shown as the mean +/- SD (unless otherwise stated), with the significance set at $p < 0.05$. The normality of the data was examined using the Shapiro-Wilk normality test, and Welch's correction was used for all t-tests where the standard deviation between groups significantly differed. Differentially expressed gene (DEG) calculations for RNA-seq were computed with DEseq2 using the Wald test, with significance denoted as a fold change > 2 and $p$-value with false discovery rate (FDR) < 0.05.

## Results

### Serum stress responses

To characterize stress responses between calves immediately following auction and transport (Stressed) and calves that had been allowed to acclimate for two months to the facility (Acclimated), we evaluated serum acute phase protein and cortisol levels. This demonstrated a >4-fold mean increase in serum haptoglobin in Stressed calves (4.44 ± 1.72 µg/ml difference, $p = 0.02$, Fig 2a), and a 9-fold mean increase in SAA (87.43 ± 17.59 µg/ml difference, $p < 0.001$, Fig 2b). LBP and cortisol levels did not differ between the two groups ($p = 0.42$, $p = 0.92$, S1 Fig).

### Health and Assessment of BRD pathogen burden by qPCR

At the time of sampling, calf mean rectal temperatures did not differ between groups ($p = 0.27$, S5 Fig). Levels of respiratory pathogens in BALF from calves were assessed by qPCR and are presented in Table 2. A full list of treatments and sampling relative to arrival are presented in S2 Table. One Stressed calf and one Acclimated calf were positive for at least

Table 1. Key cellular markers defining major immune cell populations.

| Major Cell Cluster | Up-Regulated Markers | Down-Regulated/Negative Markers |
|---|---|---|
| Macrophage | CD163, SIRPA, MERTK | CD3E, PAX5, CSF3R, TOP2A |
| Cycling Macrophage | CD163, SIRPA, TOP2A, MKI67 | CD3E, PAX5, CSF3R, |
| CD4 Tcell | CD3E, LCK, CD4 | CD163, PAX5, CSF3R, CD8 |
| CD8 Tcell | CD3E, GZMA, CD8 | CD163, PAX5, CSF3R, CD4 |
| NK Cell | NCR1, KLRB1, KLRF1 | CD163, PAX5, CSF3R, CD3E |
| NK Tcell | CD3E, CD8, KLRK1, NKG2A | CD163, PAX5, CSF3R, NCR1 |
| γδ Tcell (gd Tcell) | CD3E, WC1, RORA | CD163, PAX5, CSF3R, CD4 |
| Neutrophil | CSF3R, IFIT3, SELL, SIRPA | CD163, PAX5, CD3E, TOP2A |
| B Cell | PAX5, MS4A1, HLA-DRA, CD19 | CD163, CD3E, CSF3R, TOP2A |

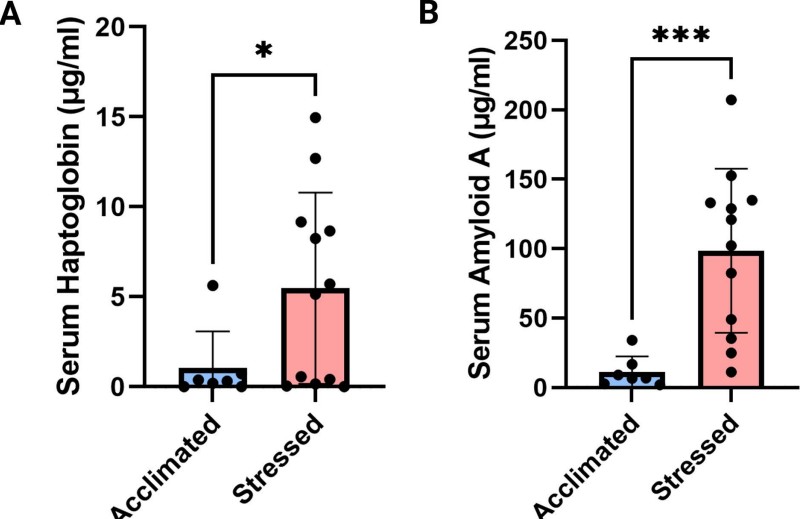

**Fig 2. Acute phase protein responses a) Serum haptoglobin levels between Stressed and Acclimated calves. b)** SAA levels between Stressed and Acclimated calves.

**Table 2. qPCR results from BALF of Stressed and Acclimated calves.**

| Calf # | Group | Bacterial Results | Viral Results | Cycle Threshold | Develop BRD in 1st 14 Days |
|---|---|---|---|---|---|
| BovLav12132 | Stressed | Neg | Neg | >35.0 | No |
| BovLav12745 | Stressed | Neg | Neg | >35.0 | No |
| BovLav105226 | Stressed | Neg | Neg | >35.0 | No |
| BovLav106915 | Stressed | Neg | Neg | >35.0 | No |
| BovLav107720 | Stressed | P. Multocida | Neg | 34.1 | Yes |
| BovLav442765 | Stressed | Neg | Neg | >35.0 | Yes |
| BovLav13014 | Stressed | Neg | Neg | >35.0 | No |
| BovLav101614 | Stressed | Neg | Neg | >35.0 | No |
| BovLav255204 | Stressed | Neg | Neg | >35.0 | No |
| BovLav253180 | Stressed | Neg | Neg | >35.0 | No |
| BovLav58778 | Stressed | Neg | Neg | >35.0 | No |
| BovLav587932 | Stressed | Neg | Neg | >35.0 | Yes |
| BovLav461618 | Acclimated | Neg | Neg | >35.0 | No |
| BovLav429617 | Acclimated | M. Bovis, H. Somni | Neg | 30.6,34.5 | No |
| BovLav420826 | Acclimated | Neg | Neg | >35.0 | No |
| BovLav463021 | Acclimated | Neg | Neg | >35.0 | No |
| BovLav465113 | Acclimated | Neg | Neg | >35.0 | No |
| BovLav742132 | Acclimated | Neg | Neg | >35.0 | No |
| BovLav466718 | Acclimated | Neg | Neg | >35.0 | No |

one respiratory pathogen at sampling, although they did not exhibit signs of respiratory infection. Three Stressed calves later went on to develop BRD signs in the first two weeks at the facility and were sampled again for pathogen screening. Interestingly, two of the three calves were negative for all respiratory pathogens by qPCR, and the other calf was positive

for BRSV (S2 Table). Stressed calves were not more likely to be positive for respiratory pathogens in BALF than Acclimated calves at first sampling (odds ratio = 0.55 [95% CI: 0.03–11.96], $p = 1$).

### Stressed calves demonstrated increased inflammatory gene expression

Differential expression analysis identified 387 differentially expressed genes between Stressed and Acclimated calves (320 upregulated in Stressed calves, 67 downregulated, Fig 3, complete list in S3 Table).

Top differentially expressed genes included inflammatory genes such as *CXCL8*, *CCL22*, *CSF1*, *CXCL2*, and inflammatory receptors *CSF3R* and *CXCR1* (Fig 4). Inflammatory cytokine signaling appeared to be a hallmark of Stressed calves relative to Acclimated, with >4-fold increases in these genes.

Hierarchical clustering was completed to assess relationships between gene expression profiles (Fig 5). This demonstrated that Acclimated calves had relatively uniform gene expression profiles, highlighting the relative homogeneity of the group. On the other hand, Stressed calves demonstrated a spectrum of gene regulation, where calves displayed varying levels of inflammatory gene expression. To better understand this variation, calves were further classified by whether they developed BRD in the first 14 days after arrival at the facility. Interestingly, while two of the calves which displayed the greatest inflammatory signatures later developed clinical BRD as identified by trained facility pen riders, differential gene expression analysis between Stressed calves who later went on to develop BRD and those who remained healthy did not

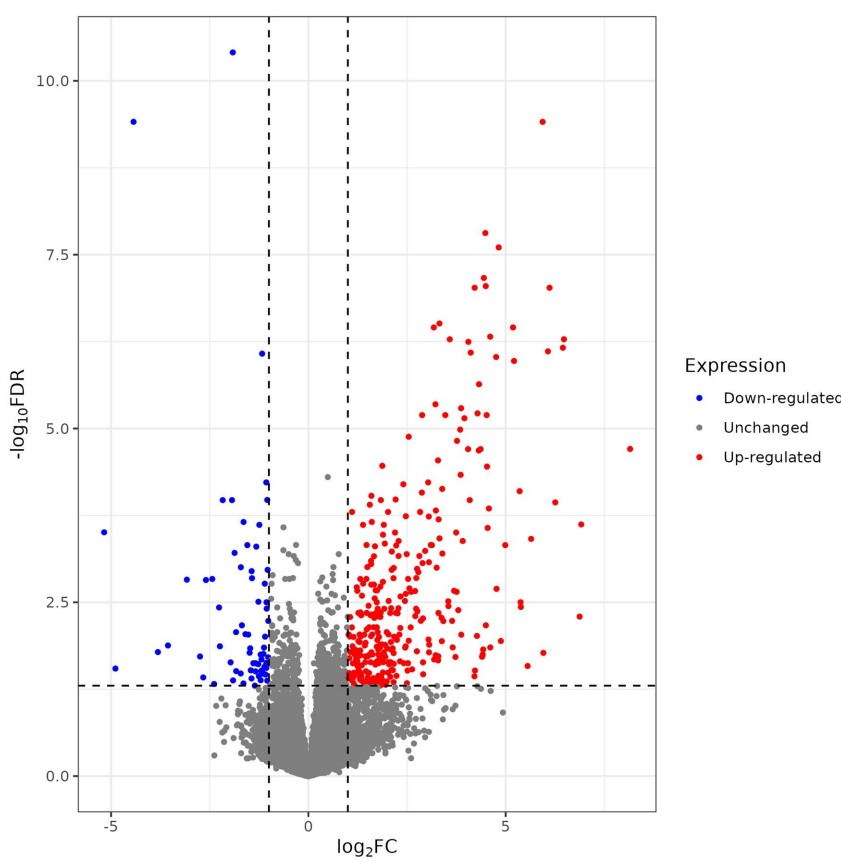

**Fig 3. Volcano plot of differentially expressed genes between Stressed and Acclimated calves (red, upregulated in Stressed calves; blue, downregulated, gray, n.s.).**

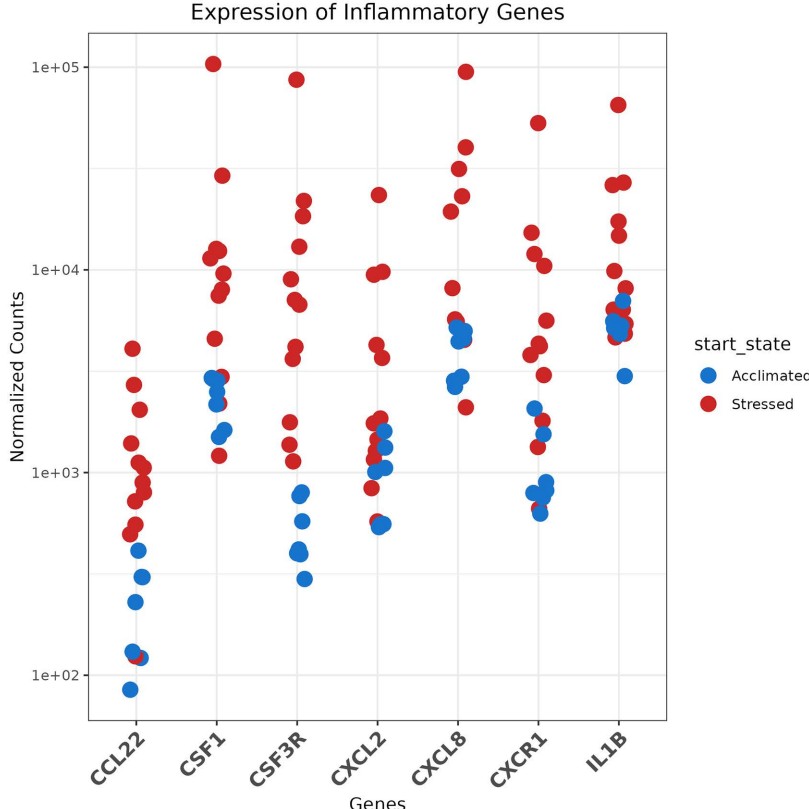

**Fig 4. Normalized counts of key upregulated genes in Stressed calves (red) vs Acclimated calves (blue).**

demonstrate appreciable differences in inflammatory signaling (S6 Fig). Therefore, inflammatory signaling appeared to predominate in Stressed calves whether they later developed BRD or not.

## Gene ontology and cell abundance analysis

To better understand this inflammatory gene expression, we conducted gene ontology analysis using GO:BP gene sets from msigdb. This demonstrated 69 upregulated pathways in Stressed calves, some of the most prominent being granulocyte migration, IL-1β and IL6 mediated signaling, and antiviral signaling. (Fig 6a, full list in S4 Table). Interestingly, this signaling was occurring even as the calves were negative for viral pathogens common to BRD pathogenesis (Table 2). Leukocyte, and more specifically, granulocyte migration was prominent in Stressed calves, so we sought to predict cell abundance proportions using CIBERSORTx [29]. A deconvolution reference was generated from annotated single-cell RNA-seq data (described in S1–S4 Figs). Using this dataset, we were able to predict cellular abundance levels. From this analysis, Stressed calves were predicted to have a four-fold increase in the level of neutrophils in their BALF ($p = 0.003$). Interestingly, lymphocyte populations were predicted to remain relatively constant. As macrophages were the most abundant cell type, the increase in neutrophil relative abundances resulted in a reciprocal 13% decrease in macrophage percentages ($p = 0.04$, Fig 6b).

## Discussion

In this study, healthy Stressed beef steers had increased inflammatory transcriptomic signatures relative to healthy Acclimated steers. In contrast to our hypothesis, differential gene expression did not demonstrate downregulation of

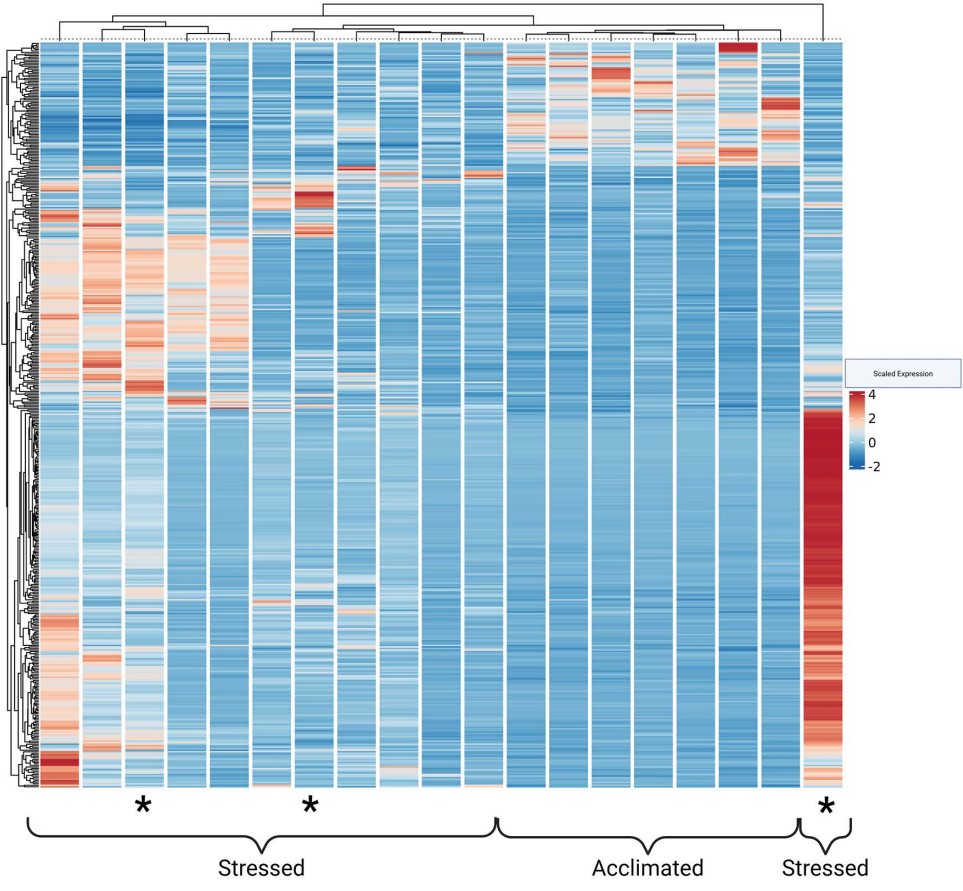

*Denotes a calf who developed BRD within 14d of arrival

**Fig 5. Heat map of differential gene expression between Stressed and Acclimated calves, highlighting how Stress calves differentiate into groups based upon levels of inflammatory gene expression.** As one Stressed calf in this representation displayed profound inflammatory signaling, Supplemental Figure 7 is included to display distribution of DEGs in calves that had similar upregulation of inflammatory genes, but not to the extent of that calf. * Denotes a calf who later went on to develop BRD.

lymphocyte signaling at the mucosal surface. Genes associated with cytotoxic lymphocyte activity and natural killer (NK) cell immunity were not differentially expressed, which is surprising given the understanding that these responses are sensitive to stress mediators such as glucocorticoids and catecholamines [30]. In contrast, neutrophilic signaling pathways were profoundly upregulated in stressed calves. Predicted neutrophil proportion increases appeared to be associated with upregulation of both interferon signaling and IL-1β signaling and were associated with mononuclear migration pathway upregulation as well. Surprisingly, this inflammatory signaling appeared to occur in the absence of appreciable pathogen colonization of the lower respiratory tract, as indicated by qPCR. It is important to note that *Bibersteinia trehalosi* was not included in the BRD qPCR panel used here, so screening for that pathogen was not conducted.

Neutrophils are recruited to BALF more strongly in stressed cattle as compared to unstressed after stimulation with LPS [14], and our data further demonstrates that even in calves that have similar pathogen burden, transit and auction stress may potentiate neutrophil recruitment to the lung. In mice, chronic stress can increase neutrophil infiltration in the lung in a glucocorticoid receptor-dependent manner [19]. Acute stress has also been associated with increased immune

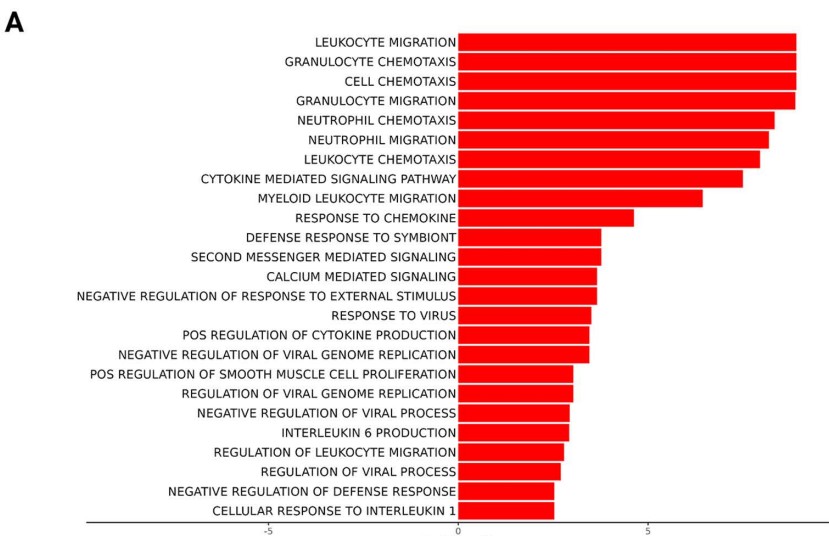

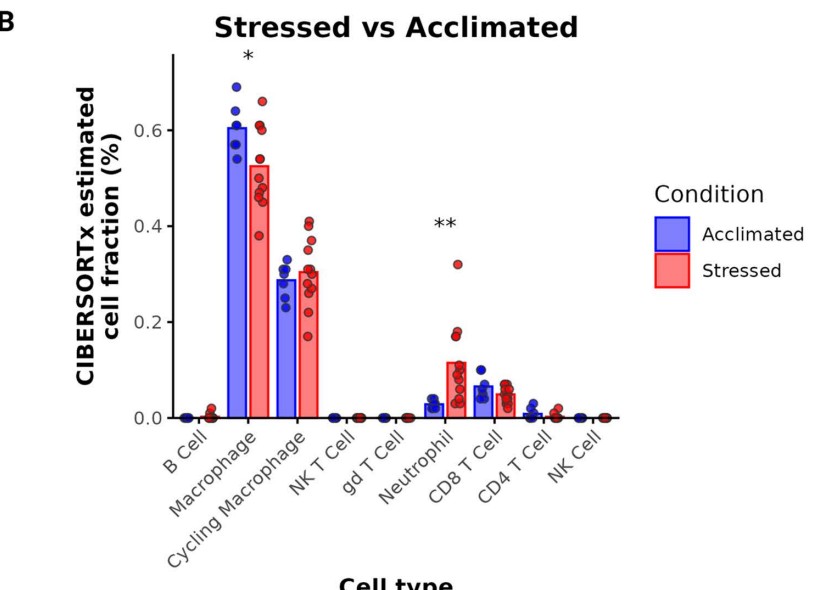

**Fig 6. Gene Ontology Analysis and Predicted BALF Immune Cell Proportions in Stressed and Acclimated calves. a)** GO:BP terms upregulated in Stressed calves as compared to acclimated calves, **b)** CIBERSORTx predicted cell proportions in Stressed (red) and Acclimated (blue) calves.

cell recruitment to sites of inflammatory stimuli in mice [21]. This study demonstrated that acute stress results in increased recruitment of many major immune cell populations, including neutrophils, T cells, and NK cells, and that the inclusion of cytokines like TNF-α in a site of inflammation shifts recruited cellular proportions to favor profound neutrophilic recruitment over lymphocytes. In cattle, this phenomenon has not been experimentally evaluated, though it could explain the neutrophilic infiltration seen here. In rats, resident immune cells like alveolar macrophages can have a more inflammatory phenotype following stress, and produce significantly more IL-1β and TNF-α upon stimulation with lipopolysaccharide (LPS) [31]. In the present study, while qPCR demonstrated that most calves were negative for BRD pathogens in the

lower respiratory tract, commensal bacteria exist in the lower airway and may stimulate neutrophilic recruitment in animals that are sensitized due to the stress of transport and commingling through an upregulation in TNF-α from alveolar macrophages. Although this has not been replicated in cattle, it is plausible that signaling in the early stages of stress from mediators such as epinephrine [32] activated macrophages to recruit innate immune cells from circulation.

It is important to note that more work is needed to characterize neutrophilic recruitment to the lung in stress. The data presented here demonstrating elevated neutrophils in BALF are predictive based upon gene expression data. Gene signatures and pathway analysis likewise provide support for cellular recruitment in stress, but future work in this area will require inclusion of functional immune characterization including flow cytometry, cytology, and cell culture.

Stressed calves in this study did not have appreciably elevated levels of serum cortisol at the time of sampling, but it is possible that peak cortisol levels had already been reached and had fallen back to baseline levels at the time of sampling, which averaged 16–18 hours after arrival at the backgrounding facility. The distance calves traveled in this study was comparable to average distances traveled by feedlot and stocker calves across the U.S [1]. Additionally, auction marketing likely contributed to stress levels, as the calves were held overnight and comingled prior to transport to the new facility. The acute phase response, as demonstrated by serum haptoglobin and SAA, appeared to be elevated in steers immediately following auction and transport. Serum haptoglobin and SAA are typically elevated in animals following stressful activities [33–35], and may be better indicators of extended stress as compared to glucocorticoids. Sympathetic mediators upregulated in stress stimulate the acute phase response and result in extended production of these proteins by the liver, which remain elevated significantly longer than cortisol following acute stress. Additionally, acute phase responses are upregulated in inflammation, specifically in response to molecules of bacterial and viral origin, including LPS. As a more sensitive indicator of potential infection [36], LBP was included in acute phase protein assessment to better ascertain if calves were experiencing low levels of LPS challenge, either from the upper respiratory tract or gastrointestinal tract. While SAA and haptoglobin levels were significantly elevated in Stressed calves, LBP levels were not significantly different between the Stressed and Acclimated calves, further supporting that these calves did not have significantly different levels of gram-negative bacteria impacting systemic immune responses.

Compared to previous research, Stressed calves appeared to have similar neutrophil proportions to calves with significant (≥1 cm) lung consolidation [37], although mean neutrophil proportions were predicted to be lower than animals with clinical BRD [38]. As even healthy calves can have significant levels of pathogen burden in the lung [39], qPCR was chosen as a means to sensitively compare pathogen burden in the lower airway. In this study, a lower percentage of healthy animals were positive for respiratory pathogens than has been reported by other studies [39,40], although one animal from each experimental group was positive for various respiratory pathogens. It is possible that greater differences could have been appreciated by evaluating upper airway levels of BRD pathogens in these calves, but importantly many of the bacterial organisms on BRD qPCR panels are considered commensals of the upper airway [5]. Aside from the one BRSV positive calf, calves who were diagnosed with BRD were not positive for respiratory pathogens by the time of second sampling, indicating that pathogen levels were either too low at that point to appreciate, or pathogens were localized to a different lung lobe than what was sampled. BALF collections were performed without visualization of the bronchus, so the lung lobe sampled could not be determined. It is notable that Stressed calves who went on to develop BRD signs in the two weeks at the facility did not have significantly greater inflammatory signaling than Stressed calves who stayed healthy. Due to the auction and commingling experienced by all calves, it would be theoretically possible for calves to arrive at the backgrounding facility already infected with a respiratory pathogen. However, qPCR and hierarchical clustering of gene expression appear to indicate that even calves who went on to develop BRD were free from excess pathogen levels in the lower respiratory tract at arrival and did not significantly differ from cattle who stayed healthy in their immune responses.

In the current study, we found inflammatory signaling to be increased in Stressed calves in the absence of elevated pathogen burden, with significant implications for further pathology. While heightened immune responses can be protective in the immediate period following weaning and transport [16,41,42], there appears to be a tipping point at which

immune stimulation in the respiratory tract leads to further pathology [42]. The principle immune cell involved in further pathology is the neutrophil, due to its capacity to effect significant tissue damage through the production of excess reactive oxygen species, proteases, and inflammatory chemokines [43]. Neutrophils provide an essential role in protection from bacterial and viral infections, and considerable research in cattle has explored maximization of neutrophil functionality in the effort to mitigate disease [5,43–45]. However, increasing evidence indicates that in cattle, pathology in stressed states is due to an excess of inflammation, which is characteristically mediated by neutrophils. Here, we propose that the recruitment of neutrophils is enhanced in Stressed cattle, even in the absence of overt pathogen challenge.

## Supporting information

**S1 Fig. Single-cell dataset (A) annotated Uniform Manifold Approximation and Projection (UMAP) of all cells included in the reference dataset, (B) characteristic markers showing distribution of feature expression on the UMAP, (C) relative expression of select marker genes across UMAP cell clusters.**
(TIF)

**S2 Fig. Heatmap displaying percent difference values between CIBERSORTx predicted cellular proportions and single-cell "ground truth" cellular proportions.**
(TIF)

**S3 Fig. Plots displaying correlation and $R^2$ values between CIBERSORTx predicted cell proportions and single-cell "ground truth" cellular proportions.**
(TIF)

**S4 Fig. Serum LBP and cortisol levels between Stressed and Acclimated calves.**
(TIF)

**S5 Fig. Rectal temperature between Stressed and Acclimated calves.**
(TIF)

**S6 Fig. Differences in transcriptomic sequencing between Stressed calves who developed BRD and those who remained healthy, including (A) volcano plot of DEGs, and (B) heatmap showing clustering of animals.**
(TIF)

**S7 Fig. Heat map between Stressed and Acclimated calves with all calves included.**
(TIF)

**S1 Table. Cluster markers (genes) for each major cell population included in the CIBERSORTx matrix.**
(CSV)

**S2 Table. Calf treatment and metadata values demonstrating time relative to arrival of sampling and treatment, as well as BRD calf sampling qPCR positives.** Ultrasound score (0–5); 0 = no lobular or lobar consolidation; 1 = diffuse comet tails; 2 = at least 1 cm$^2$ of lobular consolidation; 3 = lobar consolidation (1 lung lobe); 4 = lobar consolidation (2 lung lobes); 5 = lobar consolidation (3 or more lung lobes).
(XLSX)

**S3 Table. Complete list of differentially expressed genes (DEGs) between Stressed and Acclimated calves.**
(CSV)

**S4 Table. GO: BP terms upregulated in Stressed calves.**
(CSV)

## Acknowledgments

We would like to thank Jade Kurihara for sample processing support and laboratory training of author GJ. Additionally, Jenaye De Seve provided indispensable field support during sample collection on-farm.

## Author contributions

**Conceptualization:** Grace M. Jakes, Steven Dow, Sarah M. Raabis.

**Data curation:** Grace M. Jakes, Randy Hunter, Sarah M. Raabis.

**Formal analysis:** Grace M. Jakes, Dylan T. Ammons, Steven Dow.

**Funding acquisition:** Steven Dow, Sarah M. Raabis.

**Methodology:** Grace M. Jakes, Randy Hunter, Steven Dow, Sarah M. Raabis.

**Project administration:** Steven Dow, Sarah M. Raabis.

**Software:** Dylan T. Ammons.

**Supervision:** Sarah M. Raabis.

**Visualization:** Grace M. Jakes, Steven Dow.

**Writing – original draft:** Grace M. Jakes.

**Writing – review & editing:** Dylan T. Ammons, Randy Hunter, Steven Dow, Sarah M. Raabis.

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
