## [Decision Letter · Decision Letter 0]

19 Aug 2025

Dear Dr. Raabis,

We look forward to receiving your revised manuscript.

Kind regards,

Angel Abuelo, DVM, MRes, MSc, PhD, DABVP (Dairy), DECBHM

Academic Editor

PLOS ONE

Journal Requirements: 

2. To comply with PLOS ONE submissions requirements, in your Methods section, please provide additional information regarding the experiments involving animals and ensure you have included details on methods of anesthesia and/or analgesia.

3. Thank you for uploading your study's underlying data set. Unfortunately, the repository you have noted in your Data Availability statement does not qualify as an acceptable data repository according to PLOS's standards.

4. Please upload a copy of Supporting Information Table 2 which you refer to in your text on page 23.

Reviewers' comments:

Reviewer's Responses to Questions

**Comments to the Author**

1. Is the manuscript technically sound, and do the data support the conclusions?

Reviewer #1: Partly

Reviewer #2: Partly

2. Has the statistical analysis been performed appropriately and rigorously?

Reviewer #1: Yes

Reviewer #2: No

3. Have the authors made all data underlying the findings in their manuscript fully available?

Reviewer #1: Yes

Reviewer #2: No

4. Is the manuscript presented in an intelligible fashion and written in standard English?

Reviewer #1: Yes

Reviewer #2: Yes

Reviewer #1: This manuscript by Jakes et al. addresses an important question in bovine respiratory disease pathogenesis, how stress impacts mucosal immune responses in the absence of overt infection. The authors use a combination of BALF RNA-seq, pathogen qPCR, and serum biomarkers to interrogate this question. The study is well-designed, and the focus on lower airway responses is a strength that adds to prior work centered on systemic or peripheral responses.

Major comments:

1) The core finding that stress induces, rather than suppresses, inflammation differs somewhat from the understanding of BRD, but I think that idea is not totally novel and is consistent with prior studies in calves (e.g., Hodgson et al., Griebel’s work), many of which the authors identified. This manuscript extends that work by focusing specifically on the lower airway and linking inflammation to pathogen-independent mechanisms, demonstrated by the absence of detectable pathogens in BALF at the time of sampling. The authors should consider softening the novelty claims around increased vs. suppressed inflammation, and instead emphasize the value of this study in refining prior observations through a mucosal, pathogen-exclusion lens.

2) The authors heavily emphasize the CIBERSort cell deconvolution data in the discussion. However, this data and analysis is not clearly described and lacks validation. More detail is needed on the reference dataset used, and the markers used to differentiate these populations. Without supporting cytology, flow cytometry, or BALF cell counts, the inferred shift in neutrophils should be interpreted cautiously and some discussion should be added regarding the limitations of relying solely on computational predictions should be acknowledged. Along these lines, Figure 7 doesn't seem to add much to the discussion section.

Reviewer #2: This manuscript describes research supporting the concept that newly received cattle have increased activation of gene expression pathways related to inflammation in airways cells, in the absence of evidence of infection, and as compared to acclimated cattle. The study does provide some support for the concept that excessive inflammation, rather than immunosuppression per se, is the state that underlies BRD in newly received stocker cattle. The results merit reporting. However, the experimental design is not clear, and some needed information is missing, as described below.

Introduction

Line 83-91: There is no mention of the fact that haptoglobin, cortisol, LBP, and SAA would be measured in the blood of study cattle. What objective was being addressed by these measurements? That should be added to this section.

Materials and Methods

Where are the methods for the CIBERSORT? They need to be included.

Were the calves in the Stressed group and the calves in the Acclimated group all from the same group of cattle purchased at the same time? Or were they purchased at different times?

Did the cattle receive antimicrobial metaphylaxis (i.e. long acting antimicrobial at arrival for BRD control)? This should be explicitly stated.

The study design is not very clear. Is it true that some Stressed calves were never diagnosed with BRD after they were sampled until the end of the backgrounding period, while others were diagnosed with BRD (and sampled when they were)?

And is it true that the Acclimated calves were never diagnosed with BRD between arrival and the day they were sampled? If so, it would help improve reader understanding of the design if this was stated explicitly. Also, how long was the total backgrounding period?

Reader understanding of the experimental design could be improved by adding more detail about the experimental design to Figure 1. For example, did all Stressed steers subsequently develop BRD, and have BAL collected when they developed BRD, or only some of them? For each Stressed animal sampled that developed BRD, how many days after arrival did BRD occur?

Line 98-100: power analysis described, but what outcome was the focus of the power calculation, and how much difference between the 2 groups in this outcome would the sample size of 9 detect?

Line 123: "foley" should be capitalized, and it seems that the term "Foley catheter" is limited to the type of cuffed catheters used for urinary catheterization. It may be more correct to simply to say "A sterile cuffed bronchoalveolar lavage catheter..." instead of "..Foley cather..."

Lines 182-186: what statistical test was used to compare cortisol and acute phase proteins between groups?

Results

Given the small number of cattle included in this study, the authors should provide metadata for each animal in both the Stressed and Acclimated groups in a spreadsheet provided as supplementary data. The metadata should information such as the weight of each animal, the day relative to arrival when the BAL and blood were collected, and all times the animal was treated for BRD or any other disease, and what treatment they received (i.e. which antimicrobial)

Where is the list of all DEG? They should be provided in a spreadsheet as supplementary data.

Line 227: "...whether they developed BRD in the first week after arrival..." Do the authors mean "first 2 weeks", as indicated in Supplementary Figure 4?

Line 255: "Clade" refers to a group of organisms that appear to have evolved from the same ancestor; it does not appear to be an appropriate term as used here. The word "clade" should be replaced with a different word in the manuscript and figures; "group" is probably adequate.

Figures

Figure 1: As mentioned above, Figure 1 could be made more informative by inclusion of more information, e.g., how many cattle in each group was treated for BRD and when. It is not clear that as currently presented Figure 1 provides enough information to warrant inclusion.

Figure 5: Some of the text of the legend of Figure 5 is unclear, what does this sentence mean: "One Stressed calf displaying the most profound inflammatory signaling was removed for this representation to demonstrate the clades of inflammatory signaling"? Please revise this legend to make it more clear.

It is not very clear why the heatmap in Figure 5 is presented in the body of manuscript and the heatmap in Figure 4 was presented as supplementary data. Why not just present the heatmap in Figure 4 in the body of the manuscript?

The asterisks in Supplementary Fig 4 indicates cattle treated for BRD within 14 days of arrival; were any treated after 14 days but before the end of the backgrounding period?

Figure 7: In the image of the alveolus of the stressed animal, what do the small green dots represent?

Because the mechanism depicted in Figure 7 is entirely hypothetical, it is not clear that this figure is truly warranted.

Supplemental Figures:

Supplemental Figure 3 has two components, A and B, but these are not both described in the legend for this figure.

**Do you want your identity to be public for this peer review?** For information about this choice, including consent withdrawal, please see our Privacy Policy

Reviewer #1: No

Reviewer #2: No

---

## [Author Response · Author response to Decision Letter 1]

3 Oct 2025

Response to Editorial Comments:

Thank you for the clarification and direction on the PLOS ONE style requirements. On further review of the provided title page template, we have removed the “Full Title” and “Short Title” sections on lines 1-6 (1-2). We have also added clarification on equal contributions for each author on line 9 (5,16-17).

We have changed our figure and supplemental figure file names to better comply with the desired file naming conventions as outlined in the PLOS ONE guidelines. Each figure we have changed from “Figure 1” to “Fig1”, and the supplemental figures and tables have been changed to include the S numbering system (e.g. S1, S2, etc.).

We have changed the main sections to a 14pt bolded font, and each subsection to a 11pt bolded font instead of using the numbering convention. If this is not the desired alteration, we can also change it back based upon your feedback.

2. To comply with PLOS ONE submissions requirements, in your Methods section, please provide additional information regarding the experiments involving animals and ensure you have included details on methods of anesthesia and/or analgesia.

We appreciate the feedback on this aspect of our methods section. To address this concern, we have included our handling practices, and chute brand/design for animal restraint on lines 129-130 (123-124). Additionally, we have added verbiage in the Study Population section about our use of xylazine and efforts to reduce animal suffering on lines 119-121 (113-115). Specific doses and brands used can be found on lines 137-140 (131-134). Based on our interpretation of PLOS ONE ‘s Animal Research guidelines, we have made every effort to include desired information regarding sedation and analgesia, but if there is additional information that is desired, we can supply it as needed.

3. Thank you for uploading your study's underlying data set. Unfortunately, the repository you have noted in your Data Availability statement does not qualify as an acceptable data repository according to PLOS's standards.

Thank you for this clarification on acceptable repositories. We have elected to upload our dataset to the Dryad repository and still include the GEO accession number so that readers can access both at their discretion. The Dryad repository reviewer link has been shared elsewhere in the submission, and can also be found here: http://datadryad.org/share/LINK_NOT_FOR_PUBLICATION/tFIZElhw0_WoF02Q1mmUKXPf3N9FbTHBVUYc6RGNbNw.

The DOI link for publishing has been included in the body of the manuscript on line 225 (217), and can also be found here:

https://doi.org/10.5061/dryad.02v6wwqh6.

4. Please upload a copy of Supporting Information Table 2 which you refer to in your text on page 23.

We apologize for this oversite and appreciate you bringing this to our attention. We have uploaded this table, now as Supporting Information/Table 4, as we have added more supporting information earlier in the document.

We did not see any reviewer comments pertaining to citing other published work, but if further in the review process there is a request for other citations, we would be more than happy to address these as they arise.

Response to Reviewer 1:

Thank you for the time and effort it took to review our manuscript. Your thoughts and comments were extremely helpful in the revision of our manuscript, and we have made every effort to address each point raised in the review process. In our response to your comments, we have included your prior comment in italics, and our response in normal font immediately following that paragraph. We have sought to thoughtfully address every point raised and can provide more support or clarification to our data and revisions as needed.

1) The core finding that stress induces, rather than suppresses, inflammation differs somewhat from the understanding of BRD, but I think that idea is not totally novel and is consistent with prior studies in calves (e.g., Hodgson et al., Griebel’s work), many of which the authors identified. This manuscript extends that work by focusing specifically on the lower airway and linking inflammation to pathogen-independent mechanisms, demonstrated by the absence of detectable pathogens in BALF at the time of sampling. The authors should consider softening the novelty claims around increased vs. suppressed inflammation, and instead emphasize the value of this study in refining prior observations through a mucosal, pathogen-exclusion lens.

We appreciate the perspective in this comment and agree that our initial language may have over accentuated the novelty of our data, specifically in the context of stress. In response to this comment, we have altered our language in the abstract, introduction, and discussion sections of the manuscript to soften and redirect our commentary to focus on the mucosal host-pathogen aspect of our findings. Please see our changes at lines 43-44 (38-40), 94 (88-89), 399-409 (382-392).

2) The authors heavily emphasize the CIBERSort cell deconvolution data in the discussion. However, this data and analysis is not clearly described and lacks validation. More detail is needed on the reference dataset used, and the markers used to differentiate these populations. Without supporting cytology, flow cytometry, or BALF cell counts, the inferred shift in neutrophils should be interpreted cautiously and some discussion should be added regarding the limitations of relying solely on computational predictions should be acknowledged. Along these lines, Figure 7 doesn't seem to add much to the discussion section.

Thank you for this comment. We have added documentation detailing our methods and markers used in this CIBERSORTx analysis. Specifically, Table 1 in the body of the document has been added to detail select markers used to classify cellular populations. This does not provide an exhaustive list of the classifications, rather is intended to be similar to a flow cytometry-style panel, demonstrating upregulated and “negative” or downregulated markers. We have used both cell surface markers and transcription factors to classify the cellular populations and have included a minimum of 3 upregulated markers per cell type. A full list of the defining upregulated and downregulated markers for each cell type can be found in Supplemental Table 1. We have also visually represented the dataset using a uniform manifold approximation and projection (UMAP) image so that the readers can see the distribution and make-up of the single-cell dataset (Supplemental Figure 1a and 1b). A dot plot heat map has been included as a visual representation of the markers used for each cell type (Supplemental Figure 1c). The numbers/columns on the dot plot represent the individual cell clusters on the UMAP, so that readers may cross-reference the distribution of cell populations in relation to one another and their major markers.

Several of the calves included in the study had paired single-cell and bulk sequencing samples, so we completed a difference from single-cell “ground truth” analysis on our CIBERSORTx predicted values for each cell type. We completed this analysis for both of the different batch correction methods evaluated and have included that data in Supplemental Figure 2. We also conducted a correlation analysis and have included the visual representation of this analysis as well as the R2 values in Supplemental Figure 3.

A detailed discussion of our CIBERSORTx methodology can be found in the Methods section, lines 197-222 (189-214). Please let us know if additional support or clarification of the data is needed.

We have added a paragraph discussing the limitations of in silico predictions of cell population proportions on lines 347-351 (336-340) in the Discussion section, and upon further reflection, have elected to remove primary Figure 7 from the manuscript.

Response to Reviewer 2:

Thank you for the time and effort it took to review our manuscript. Your thoughts and comments were extremely helpful in the revision of our manuscript, and we have made every effort to address each point raised in the review process. In our response to your comments, we have included your prior comment in italics, and our response in normal font immediately following that paragraph. We have sought to thoughtfully address every point raised and can provide more support or clarification to our data and revisions as needed.

Introduction Line 83-91: There is no mention of the fact that haptoglobin, cortisol, LBP, and SAA would be measured in the blood of study cattle. What objective was being addressed by these measurements? That should be added to this section.

We appreciate this feedback. We sought to use these markers to demonstrate that calves in this study were truly stressed, or at least responding systemically to the transit and auction stressors that they had experienced. As we were not in control of exactly when the calves would arrive and be sampled (beyond within the first 18-24 hours after arrival), we chose multiple stress markers that covered early and later stress responses. We have updated the manuscript to reflect this in lines 87-88 (82-83).

Materials and Methods Where are the methods for the CIBERSORT? They need to be included.

Thank you for this clarifying request. We have included detailed methodology on lines 197-222 (189-214). We also wanted to respond with some explanation for each supplemental figure and table added for the CIBERSORTx methods:

Table 1 in the body of the document has been added to detail select markers used to classify cellular populations. This does not provide an exhaustive list of the classifications, rather is intended to be similar to a flow cytometry-style panel, demonstrating upregulated and “negative” or downregulated markers. We have used both cell surface markers and transcription factors to classify the cellular populations and have included a minimum of 3 upregulated markers per cell type. A full list of the defining upregulated and downregulated markers for each cell type can be found in Supplemental Table 1. We have also visually represented the dataset using a uniform manifold approximation and projection (UMAP) image so that the readers can see the distribution and make-up of the single-cell dataset (Supplemental Figure 1a and 1b). A dot plot heat map has been included as a visual representation of the markers used for each cell type (Supplemental Figure 1c). The numbers/columns on the dot plot represent the individual cell clusters on the UMAP, so that readers may cross-reference the distribution of cell populations in relation to one another and their major markers.

Several of the calves included in the study had paired single-cell and bulk sequencing samples, so we completed a difference from single-cell “ground truth” analysis on our CIBERSORTx predicted values for each cell type. We completed this analysis for both of the different batch correction methods evaluated and have included that data in Supplemental Figure 2. We also conducted a correlation analysis and have included the visual representation of this analysis as well as the R2 values in Supplemental Figure 3.

Were the calves in the Stressed group and the calves in the Acclimated group all from the same group of cattle purchased at the same time? Or were they purchased at different times?

Thank you for this question. We randomly sampled cattle in batches of 3-4 from different shipments throughout the purchasing season. This was done to ideally get the most randomized and representative sample from the backgrounding period, not biased by calf source. We have updated the manuscript to reflect this on line 103-104 (97-98).

Did the cattle receive antimicrobial metaphylaxis (i.e. long acting antimicrobial at arrival for BRD control)? This should be explicitly stated.

Thank you for this question. The calves did not receive any metaphylaxis, nor did any calf that arrived or was put out to pasture at the same time as the enrolled calves. We have added this information on line 112 (106).

The study design is not very clear. Is it true that some Stressed calves were never diagnosed with BRD after they were sampled until the end of the backgrounding period, while others were diagnosed with BRD (and sampled when they were)?And is it true that the Acclimated calves were never diagnosed with BRD between arrival and the day they were sampled? If so, it would help improve reader understanding of the design if this was stated explicitly. Also, how long was the total backgrounding period?

We appreciate this feedback and apologize for the lack of clarity. Yes, 7 of the Stressed calves were never diagnosed with BRD throughout the backgrounding period. Calves that developed BRD within 14 days of arrival were sampled at the time they were pulled (before antibiotic administration). Two calves developed BRD ~6 weeks after arrival and were not sampled because the backgrounder could not coordinate with our sampling over the Thanksgiving holiday. We have updated the manuscript to address this briefly on lines 158-160 (151-152).

Acclimated calves were never diagnosed with BRD between arrival and when they were sampled (and through the entire backgrounding period). We have updated the manuscript to reflect this on line 149-152 (142-145).

Reader understanding of the experimental design could be improved by adding more detail about the experimental design to Figure 1. For example, did all Stressed steers subsequently develop BRD, and have BAL collected when they developed BRD, or only some of them? For each Stressed animal sampled that developed BRD, how many days after arrival did BRD occur?

Thank you for this suggestion. We have substantially changed Figure 1 to include the timeline of sampling/the backgrounding period, and to include the number of Stressed calves who developed BRD and were sampled again. We included the date range for when we sampled calves who had developed BRD, and the specific number of days can be found in Supplemental Table 2.

Line 98-100: power analysis described, but what outcome was the focus of the power calculation, and how much difference between the 2 groups in this outcome would the sample size of 9 detect?

We appreciate this clarifying question. We based our power analysis on expected differences in acute-phase proteins and cortisol, as these were our validation markers for stress. We derived our expected difference between the two groups from recently published literature (Beenken et al., 2021 in our references, among others) and anticipated that each of the stress mediators would be elevated in the study animals. We were incorrect about cortisol and LBP, but we addressed theories on why this could be (time of sampling, possible lack of LPS stimulation) in the Discussion. The details regarding specific values for the difference we were expecting to see can be found on lines 103-109 (98-104).

Line 123: "foley" should be capitalized, and it seems that the term "Foley catheter" is limited to the type of cuffed catheters used for urinary catheterization. It may be more correct to simply to say "A sterile cuffed bronchoalveolar lavage catheter..." instead of "..Foley cather..."

Thank

---

## [Decision Letter · Decision Letter 1]

26 Nov 2025

Dear Dr. Raabis,

We look forward to receiving your revised manuscript.

Kind regards,

Angel Abuelo, DVM, MRes, MSc, PhD, DABVP (Dairy), DECBHM

Academic Editor

PLOS ONE

Journal Requirements:

Reviewers' comments:

Reviewer's Responses to Questions

**Comments to the Author**

Reviewer #1: All comments have been addressed

Reviewer #2: (No Response)

2. Is the manuscript technically sound, and do the data support the conclusions?

Reviewer #1: Yes

Reviewer #2: Yes

3. Has the statistical analysis been performed appropriately and rigorously?

Reviewer #1: Yes

Reviewer #2: Yes

4. Have the authors made all data underlying the findings in their manuscript fully available?

Reviewer #1: Yes

Reviewer #2: Yes

5. Is the manuscript presented in an intelligible fashion and written in standard English?

Reviewer #1: Yes

Reviewer #2: Yes

Reviewer #1: (No Response)

Reviewer #2: The authors have done a very good job addressing the previous review. Two small recommendations:

Line 225: Supplemental Figure 2 should be changed to Supplemental Figure 5

Supplemental Table 2: readers may be interested to know which calves had a fever and which calves had consolidated lung found on ultrasound; that information could be added to Supplemental Table 2.

**Do you want your identity to be public for this peer review?** For information about this choice, including consent withdrawal, please see our Privacy Policy

Reviewer #1: No

Reviewer #2: No

---

## [Author Response · Author response to Decision Letter 2]

30 Dec 2025

Dear Dr. Abuelo,

Thank you for your continued consideration of our manuscript entitled, “Transport stress induced paradoxical increases in airway inflammatory responses in beef stocker cattle.” We sincerely appreciate the feedback included in this request for edits. We have sought to address these points individually and to the best of our ability and will appreciate any additional feedback should it arise.

Similar to our first round of edits, any in-text changes will be cited by line number in the “Revised Manuscript with Track Changes” document, and then in parenthesis for the Manuscript document (e.g. lines 115-117 [105-107]). We are only aware of one request for an in-text edit, as requested by Reviewer 2. We have made a few minor edits to our reference list based upon closer review of the style guidelines and have included specifics on those edits in the Response to Editorial comments section. Included in the editorial comments was a request to ensure that no retracted articles were included in our list of references. We are unaware of any retracted articles included in our list of references but would make every effort to remove and replace such a reference should they exist.

Please see our specific responses to editorial and reviewer comments in the following sections. Should any response not satisfactorily address a point raised by the editors or reviewers, we would be more than happy to revisit any point more thoroughly if indicated.

Response to Editorial Comments:

1. If the reviewer comments include a recommendation to cite specific previously published works, please review and evaluate these publications to determine whether they are relevant and should be cited. There is no requirement to cite these works unless the editor has indicated otherwise. Please review your reference list to ensure that it is complete and correct. If you have cited papers that have been retracted, please include the rationale for doing so in the manuscript text, or remove these references and replace them with relevant current references. Any changes to the reference list should be mentioned in the rebuttal letter that accompanies your revised manuscript. If you need to cite a retracted article, indicate the article’s retracted status in the References list and also include a citation and full reference for the retraction notice.

Thank you for this opportunity to review our submitted references. Upon further review, we have edited the journal names to include the NLM title abbreviation rather than the full journal name for each entry. We apologize for our incorrect interpretation of the style guidelines in our initial submission. We have also slightly edited the USDA report entries to better meet the citation guidelines. Finally, the pwr package citation has been edited to link to the CRAN repository rather than the author’s GitHub page, as based upon further review we believe that this may align more correctly with Vancouver citation guidelines. There was one citation (Nyhlén et al., 2000) where the citation was inadvertently included in caps rather than sentence case, and that has been corrected. We regret this oversite.

We have made every effort to review our list of references to make sure that no retracted articles are included in our reference list. We are unaware of any article(s) that have been retracted and would be more than willing to remove any reference that is retracted should that be the case.

We have not received any requests from reviewers to cite additional work, but if there are gaps in our discussion or introduction that should be filled, we would be more than happy to review any relevant literature not already included if indicated.

Response to Reviewer 1:

Thank you for your careful review of our revised manuscript. We appreciate your effort in the review process and would welcome any further comments should they arise.

Response to Reviewer 2:

Thank you for your careful consideration of this revised manuscript draft. We appreciate your comments and have done our best to address each one.

Thank you for catching the mislabeling/reference to Supplemental Figure 5. We have corrected this in line 245 (245) of the Manuscript.

We also appreciate your suggestion of including the ultrasound score and rectal temperature at the second BAL sampling, when calves were diagnosed with BRD by pen-riders, in our supplemental data. This has been included in Supplemental Table 2 (S2 Table) under columns “Rectal Temperature at Second BAL (degrees C)” and “Lung Ultrasound Score at Second BAL.” It is important to note that the ultrasound score was 0 on two of the three calves, but based upon rectal temperature, behavior, upper respiratory signs, and caretaker best-judgement, these calves were diagnosed with BRD and were treated. The identification and treatment of these calves prior to the development of severe lung lesions is likely a testament to their acute development of signs and timely identification by pen riders before severe pulmonary disease could develop. Additionally, ultrasound examination was limited in this study, as we were unable to visualize the right cranial lung lobe consistently due to the logistics of the chute design (see lines 129-130). Therefore, lesions may have been present and not detected by ultrasound exam.

We want to reiterate our appreciation of your time and effort in reviewing our manuscript. Should you identify any further changes in the review process, we would be more than happy to address them as they arise.

Concluding Remarks

We want to conclude by reiterating our appreciation for the editorial and reviewer comments. They have substantially advanced the quality of our submission.

Thank you again for your consideration,

Grace Jakes (1st author)

Sarah Raabis (co-corresponding author)

---

## [Editor Report · Decision Letter 2]

4 Jan 2026

Transport stress induces paradoxical increases in airway inflammatory responses in beef stocker cattle

PONE-D-25-35522R2

Dear Dr. Raabis,

We’re pleased to inform you that your manuscript has been judged scientifically suitable for publication and will be formally accepted for publication once it meets all outstanding technical requirements.

Kind regards,

Angel Abuelo, DVM, MRes, MSc, PhD, DABVP (Dairy), DECBHM

Academic Editor

PLOS One
---

## [Editor Report · Acceptance letter]

PONE-D-25-35522R2

PLOS One

Dear Dr. Raabis,

I'm pleased to inform you that your manuscript has been deemed suitable for publication in PLOS One. Congratulations! Your manuscript is now being handed over to our production team.

Kind regards,

on behalf of

Dr. Angel Abuelo

Academic Editor

PLOS One